# Effects of Group IVA Elements on the Electrical Response of a Ge_2_Se_3_-Based Optically Gated Transistor

**DOI:** 10.3390/mi15081000

**Published:** 2024-08-01

**Authors:** Md Faisal Kabir, Kristy A. Campbell

**Affiliations:** Electrical and Computer Engineering, Boise State University, Boise, ID 83725, USA

**Keywords:** selector, phototransistor, optically gated transistor, OGT, chalcogenide

## Abstract

The optically gated transistor (OGT) has been previously demonstrated as a viable selector device for memristor devices, and may enable optical addressing within cross-point arrays. The OGT current–voltage response is similar to a MOSFET device, with light activating the gate instead of voltage. The OGT also provides a naturally built-in compliance current for a series resistive memory element, determined by the incident light intensity on the gate, thus keeping the integrated periphery circuitry size and complexity to a minimum for a memory array. The OGT gate comprises an amorphous Ge_2_Se_3_ material that can readily be doped with other elements to alter the transistor’s electrical properties. In this work, we explore the operation of the OGT when the Ge_2_Se_3_ gate material is doped with the Group IVA elements C, Si, Sn, and Pb. The dopant atoms provide changes to the optical and electrical properties that allow key electrical properties such as the dark current, photocurrent, switching speed, and threshold voltage to be tuned.

## 1. Introduction

Resistive random-access memory (ReRAM) devices, including memristor devices, that exhibit a continuous range of states have been investigated for applications such as analog computing [1] and artificial intelligence systems [2]. A typical computing architecture uses a cross-point array structure [3,4], where a memory element is accessed by a particular wordline (bottom electrode) or bitline (top electrode) in a simple cross-point array configuration. However, when a ReRAM array is operated this way, given the resistive nature of each memory element, these architectures are prone to inaccurate data reporting due to sneak path currents [5,6,7]. These stray currents can add to the actual addressed memory cell current, thus causing the state of the cell to be incorrectly read. To reduce this issue, accessing a memristor in a cross-point memory array would benefit from the use of a selector device or access transistor to gate each memory bit and thus to reduce or prevent the sneak path current [3].

Many selector devices use Se-based materials. These usually have three electrical contact terminals, and are controlled typically with electric potentials [8,9,10,11,12,13,14]. In recent years, a large number of phototransistor devices that use many different material types have emerged [14,15,16,17,18,19,20,21,22,23,24,25,26,27,28,29,30,31,32,33,34,35], most of which have three terminals and some of which have two [23,24]. The optically gated transistor (OGT) studied in this work is a two-terminal device comprising a Ge-Se-based gate material [13]. Other existing two-terminal phototransistors include various optoelectronic synaptic transistors such as those comprising inorganic materials such as InGaZnOx (IGZO) [18,25] and InZnO (INZO)) [26], which show promise for bioinspired computing. The IGZO [25] and INZO [26] phototransistors use a Si/SiO_2_ substrate similarly to the OGT, and show promise in bioinspired applications such as synapses and exhibit memory retention. In contrast, the OGT does not retain a memory state, but by design, functions as a switch to allow access to a memristor bit in an array.

The ability to use light intensity to program memristor devices in a cross-point array [20,23] is a promising method for achieving low-sneak-path currents, and higher-density architectures for neuromorphic computing. The OGT was initially created for use as a selector device for ReRAM [13], and specifically for use with the existing commercially available memristor devices that are based on a-Ge_2_Se_3_ material, which operate under a Self-Directed Channel (SDC) memristor mechanism [36,37,38,39]. To provide processing compatibility along with the potential to streamline an in-situ-fabricated combination memristor/selector device, the OGT devices were designed to contain the same active material as the SDC memristor that comprises a Ge-rich amorphous material, a-Ge_2_Se_3_. In the class of memristors that operate within the SDC mechanism, the Ge-Ge homopolar bonds are critical for channel formation and are known to be unstable in the presence of dopants, a property which is beneficial for reliable resistive RAM operation. It is this bond that is most likely to participate in bonding with dopants, and to aid in modulating the phototransistor’s optical and electrical properties.

The OGT exhibits current–voltage (I-V) curves that have an appearance similar to three terminal transistors, such as the common metal oxide semiconductor field effect transistor (MOSFET), as shown in Figure 1, and other phototransistor devices [27,33]. In these three terminal devices, the channel current between the source and drain electrodes is controlled by the voltage applied to the gate electrode. In the OGT, the channel current is controlled by the light applied to the gate, and is influenced by both the light intensity and the wavelength.

The ability to change the selector device’s operational properties to suit differing requirements, such as a low current, higher threshold voltage, or high speed, for example, is desirable for flexible circuit design and applications. This is even more desirable if a simple doping alteration of the same material system as the memristor can produce this change. In this work, we explore the operation of the OGT when the a-Ge_2_Se_3_ gate material is doped with the Group IVA elements C, Si, Sn, and Pb. These were selected since they are in the same periodic table group as Ge [40], and allow the exploration of the influence of the Ge-Se and Ge-Ge bond changes on the OGT switching response and the photocurrent that ultimately determines the compliance current for the memory array element. The desired OGT device properties depend entirely on the required application; therefore, the results of the influence of each type of dopant on OGT operation are presented to provide a given range of possible parameter changes, and to demonstrate the viability of simple doping on the tunability of the operational properties.

## 2. Materials and Methods

The equipment and methods for fabricating and testing the OGTs are described here. All data analysis and curve fitting was performed using IGOR Pro 9, WaveMetrics, Inc. (Portland, OR, USA) [41].

### 2.1. Fabrication of OGT Devices

The OGT structure consists of a blanket layer of M + Ge_2_Se_3_, (M = none, C, Si, Sn, Pb) sputtered onto a Si wafer with native oxide present. A shadow mask is used to generate many OGT structures over the wafer surface. Figure 2a shows an image of a completed series of OGT devices with different separation lengths, ranging from 5 mm to 1 mm, between the source and drain electrodes. In the work presented here, the 2 mm separation devices are used for electrical measurements. The OGT device fabrication steps and structure with the M + Ge_2_Se_3_ layer are in Figure 2b.

To fabricate the OGTs, three different substrates were used for the OGT thin film gate layer deposition: (1) for OGT devices, a p-Si with a native SiO_2_ layer (1–100 Ω-cm, Microsil, LLC Silicon Services, Nampa, ID, USA); (2) for the optical transmission spectra, high-purity fused quartz (TED PELLA, INC., Redding, CA, USA, Product No: 26013, 25.4 × 25.4 × 1 mm thick); and (3) for Raman measurements, wafers specially made with Cr (250 Å)/W (500 Å)/Si_3_N_4_ (500 Å) (order is from bottom towards surface) deposited on p-Si, purchased from Encompass Services. Each substrate was cleaned with acetone and allowed to air dry before the deposition of the OGT gate layer.

An AJA International ATC Orion 5 UHV Magnetron sputtering system and a Ge_2_Se_3_ target from Processed Materials (2″ dia. × 0.125″ thick, RF sputter) were used to sputter the OGT gate layer (s). To incorporate the Group IVA elements, the OGT gate was deposited by in situ sequential sputtering: (1) 100 Å Ge_2_Se_3_ (RF sputtered, 100 s); (2) co-sputtered M + Ge_2_Se_3_, (M = C, Si, Sn, Pb) (10 s); (3) a repeat of steps (1) and (2); and (4) a final sputter step for Ge_2_Se_3_ (100 s), achieving a total thickness of ~320 Å. The sputter process for these thin amorphous layers causes the in situ mixing of the M element into the overall film, creating an effectively single gate layer. The undoped Ge_2_Se_3_ device gate layer was deposited with one continuous 300 s sputter deposition step. The C target was from Process Materials (2″ dia. × 0.125″ thick, DC sputter); the Si was from AJA International (2″ dia. × 0.188″ thick, RF sputter); the Sn target was from AJA International (2″ dia. × 0.250″ thick); and the Pb target was from Kurt J. Lesker (2″ dia. × 0.125″ thick, DC sputter). For the undoped OGT, the active layer was a single ~320 Å Ge_2_Se_3_ film. W electrodes (350 Å) were sputter deposited on the surface using a shadow mask (OSH Stencils) to define the source and drain electrodes. This process is outlined in Figure 2b.

### 2.2. Material Characterization

#### 2.2.1. Raman Spectroscopy

A Renishaw inVia confocal Raman Microscope system with a 1 μm beam spot size and a 785 nm laser diode was used to obtain the Raman spectra of the films in order to measure the total OGT gate stack film thickness, stoichiometry and bonding environment in the material, and the incorporation of M in the M + Ge_2_Se_3_ sputtered layers. SEM imaging (Appendix A) also shows the homogeneity of the resultant gate material. The specialty Raman substrate was used to optimize the intensity of the Raman signals by providing a block against underlying large Si Raman signals and to provide a factor of two increase in the signal-to-noise ratio through radiation reflection off of the W layer under the nitride layer, and to allow for film thickness measurement against a laboratory Ge_2_Se_3_ thickness calibration standard.

#### 2.2.2. UV-Vis Spectroscopy

Transmission (%Tr) and reflection (%R) spectra as a function of the wavelength of the OGT films were collected using an n&k Technologies 1280 broadband UV-Vis spectrometer. The optical energy gap ($E_{0}$) was estimated from the optical transmission spectra by using the Tauc method [42].

### 2.3. Electrical Testing

LEDs ranging in wavelength from 385 to 1200 nm were used to drive the OGT gate. The vendor and part number for each LED are listed in Table 1. The power density given is measured at the detector, with a 0.71 cm^2^ area.

A summary of the electrical measurements performed is provided in Table 2. The experimental details for each electrical measurement follow.

#### 2.3.1. I-V Measurements

The circuit diagram for both the I-V sweep and LED pulse tests is shown in Figure 3. The I-V measurements were performed using a micromanipulator 6200 microprobe station with Micromanipulator 7B tungsten probe tips, and an HP4156A semiconductor parameter analyzer. During I-V sweep measurements, the force voltage was applied to the drain electrode with the source electrode maintained at ground potential. In all samples except the Pb-doped one, the drain was connected to a +V with respect to the source. The Pb-doped OGT was unique in that it required a negative potential on the drain electrode in order to operate, indicating a reversal in the majority carrier compared to the other device types (analogous to the PMOS and NMOS transistors). All I-V curve measurements used the double sweep option and were measured under dark and light conditions.

A Thorlabs PM16-121 USB Power Meter with a Si detector area limited to a circular size of 0.71 cm^2^ was used to measure the LED light power density. An LED light source vertically incident to the OGT surface and positioned closer to the drain was used for the OGT illumination in all measurements. Since the LED light is partially obscured by the drain electrode, an estimate of the area obscured was removed from the effective Si detector area, reducing the light-exposed gate area to 0.61 cm^2^.

An MC-Systems temperature-controlled wafer hot chuck was used during the temperature-dependent I-V sweep measurements, over a temperature range of 20 to 140 °C in 20 °C increments. Samples were allowed to equilibrate for 30 min once the set point temperature had been reached, before any measurements were taken. Heating occurred on the OGT wafer bottom surface, with the top exposed to ambient air in a stagnant environment. The temperature cycling measurements were independently performed on the samples three times in order to measure any functional or detrimental effects of temperature exposure.

#### 2.3.2. Pulsed Light Tests

In the pulsed light tests (solid line connections, Figure 3), a 470 nm LED was supplied a train of square pulses (by a Digilent AD2) to cycle the light on the OGT gate ON and OFF. The pulse train was set at a frequency of 100 Hz, a 1% duty cycle, and had a resultant ON pulse width of 0.1 ms. The LED was placed near the drain electrode. An E3631A power supply provided +7 V DC to the drain electrode to bias the OGT in the saturation region. The voltage across a 1 kΩ resistor connected between the source electrode and ground was monitored by using the Digilent AD2 scope input channel with a 1 MΩ input impedance, in order to calculate the current through the OGT.

### 2.4. Electrical Data Analysis Methods

The electrical characterization analysis for the measurements listed in Table 2 is described in this section.

#### 2.4.1. I-V Measurements

The I-V measurements consist of a current measurement (I_DS_) through the OGT while the voltage between the drain and source electrodes, V_DS_, is varied. Measurements are made under dark and illuminated conditions. Measurements were also taken at temperatures ranging from 20 to 140 °C, in 20 °C increments. The photocurrent, I_ph_, is the difference between the current measured during illumination and that measured in the dark [43], as shown in Equation (1).
(1)$$I_{ph}\left(A\right)=I_{Light}-I_{Dark}$$

##### Power Density

The power density is measured by positioning the LED beam at a height above the OGT sample corresponding to the power meter detector area, which is 0.71 cm^2^. The power incident on the device is then determined by multiplying the measured power density by the illuminated OGT area. The effective OGT gate illumination area was estimated at 0.61 cm^2^ since the drain electrode is partially illuminated.

##### Responsivity

The responsivity [43], R, measures the electrical current output in response to an incident optical input. This is measured by collecting I-V curves under illumination using a range of LED wavelengths and power densities. R is calculated using Equation (2), where the area, A, is considered the illuminated active area, 0.61 cm^2^; I_ph_ is the photocurrent; and P is the incident LED light power density (given in Table 1).
(2)$$R=\frac{I_{ph}(A)}{P\left(\frac{W}{{cm}^{2}}\right)*A(cm^{2})}(A/W)$$

##### Conduction

In the simplest estimate, the room-temperature conductivity is frequently modeled by Equation (3). However, this estimate does not provide detail on all of the specific mechanisms that give rise to conduction in amorphous materials such as the OGT gate (a-Ge_2_Se_3_).
(3)$$\sigma =\frac{I}{V}\frac{L}{A}.$$

Instead, Equation (3) is used to estimate a conductivity value at V = |V_DS_| = 7 V, with L and A set equal to 1 since these parameters are not accurately known, with the current, I, measured at that voltage.

The extended states conduction mechanism is dominant at room temperature and above for amorphous materials [44]. It is modeled by a first-order mechanism using the Arrhenius Equation (4):(4)$$\sigma =\sigma_{0}\exp\left(\frac{-E_{a}}{kT}\right).$$
where σ_0_ is the pre-exponential factor, k is Boltzmann’s constant, and E_a_ is the extended states conduction activation energy. Plots of Ln(σ) vs. 1/(kT) were used to estimate E_a_ from the slope of the linear fit.

#### 2.4.2. LED Pulse Testing Analysis Methods

To measure the response time of the OGT, the LED is driven with electrical pulses to produce light pulses. The pulsed light incident on the gate allows the rise and fall times of the OGT electrical response to be measured.

The rising and falling edges of the OGT device’s response to an LED light pulse were modeled using either a single or double exponential function, as shown in Equation (5) or (6),
(5)$$y=y_{0}+A\exp\left\{-\frac{\left(x-x_{0}\right)}{\tau }\right\}\;\mathrm{and}$$
(6)$$y=y_{0}+A_{1}\exp\left\{-\frac{\left(x-x_{0}\right)}{\tau_{1}}\right\}+A_{2}\exp\left\{-\frac{\left(x-x_{0}\right)}{\tau_{2}}\right\},$$
where the A’s and τ’s are the pre-exponential factors and time constants, respectively.

## 3. Results

### 3.1. Material Characterization

#### 3.1.1. Raman Spectroscopy

The Raman spectra for each OGT thin film sample type are shown in Figure 4. The spectra are each normalized with respect to the Ge-Se corner-sharing (CS) tetrahedral peak of the undoped Ge_2_Se_3_ sample, which is located at 200 cm^−1^ in Figure 4. The peak at ~175 cm^−1^ and the broad peak between 250 and 300 cm^−1^ (marked by *) correspond to Ge-Ge bonds [45]. The peak at 215 cm^−1^ is only slightly visible in these samples, but prominent in Se-rich samples (Ge_x_Se_100-x_ with x ≤ 33), and corresponds to Ge-Se edge-sharing (ES) tetrahedra. The dashed line at ~262 cm^−1^, with no visible peak in the undoped sample, would correspond to Se-Se bonds if present, as can be found in Se-rich glasses. In the Ge-rich samples, this Se-Se peak is absent, since the concentration of Ge atoms is too high, leading to the under coordination of the Ge atoms, the formation of Ge-Ge bonds, and the lack of formation of Se-Se bonds.

The Raman spectra show that each sample is amorphous, and that each sample has been doped. The lower-atomic-weight elements, C, Si, and Ge, exhibit very similar Raman peaks corresponding to the Ge-Se (CS) and Ge-Ge bonds, which is more apparent when viewing the overlapped normalized spectra, as shown in Figure 4b. However, the Si-doped sample exhibits a broadening on the lower wavenumber side of the 175 cm^−1^ Ge-Ge peak, likely due to the formation of a peak corresponding to Ge-Si bonds. There also appears to be a small reduction in the intensity of the broad (250–300 cm^−1^) peak in the Si-doped sample compared to the undoped sample. Interestingly, in the Sn-doped sample, a very large intensity peak at 165 cm^−1^ dominates as a shoulder peak of the 175 cm^−1^ Ge-Ge peak, and concurrently, the broad Ge-Ge peak between 250 and 300 cm^−1^ appears to have been modified to a more Gaussian-shaped peak centered at 277 cm^−1^. The peak within the 250–300 cm^−1^ range has also been observed in GeSn alloys and assigned to a Ge-Sn bond [46,47]. Additionally, the peak at 165 cm^−1^ is expected for Sn-doped Ge_2_Se_3_ since it has also been observed in the Raman spectra of bulk glass Ge_2_Se_3_ samples doped with 3% Sn prepared quantitatively using a melt-quench technique (Appendix A). The Pb-doped sample also shows a large peak on the shoulder of the Ge-Ge peak, shifted to an even lower value (159 cm^−1^) than the Sn sample.

#### 3.1.2. UV-Vis Analysis

The %Tr data are given in Figure 5a. An optical energy gap, $E_{0}$, is estimated from a Tauc plot, Figure 5b, where a dashed line (corresponding to a linear fit to the mid-energy linear region) crosses the 0 axis. Since the films are amorphous, it is not possible to correlate this optical gap with a band gap, given that there are no energy bands in the amorphous material. It is also not possible to use this value to determine if the electron transition is from the valence to extended states, or from tail states or localized states [44]. However, this optical gap derived from a Tauc plot is a frequently used metric with amorphous materials and allows comparisons between researchers and material systems.

The extrapolated values of $E_{0}$ are listed in Table 3. $E_{0}$ for the Sn and Pb samples is much lower than for the other samples, and is likely due to an increase in the density of states near the conduction extended states [48,49,50,51], corresponding to an increase in the concentration of the metallic dopants compared to the other samples. However, for both the Sn and Pb-doped samples, it is expected that the concentration of metal dopant is larger than 5 at. % when considering the Raman spectra and the extrapolated $E_{0}$ values. For the Sn-doped sample, a comparison of the thin film OGT sample Raman spectra with a bulk sample containing 3 at. % Sn (Appendix A) shows that the peak at 165 cm^−1^ is much smaller than it is in the thin film sample. This peak grows in correlation with the increased % of Sn added to the sample. Estimating that the peak grows linearly with the concentration of Sn, the thin film OGT sample likely contains between 8–9 at.% Sn. At these estimated optical gaps, the Sn and Pb-doped OGTs should be more conductive in dark conditions than the C, Si, and undoped samples. The decrease in $E_{0}$ due to Pb incorporation in Ge-Se-based materials is also reported by Mishra et al. [52]. Reflection measurements were also made and a simple approximation of absorption was calculated, using the relationship Transmission + Reflection + Absorption = 1 (Appendix A). It is clear that the absorption of the Sn and Pb samples is greater than the other samples over the wavelength range of 300 to 1000 nm, supporting the decrease in optical gap.

The optical gaps for the C- and Si-doped samples are slightly smaller than the undoped sample, likely due to an increase in localized tail states in the doped samples.

### 3.2. Electrical Characterization

#### 3.2.1. Temperature Dependent I-V

Since amorphous materials are well known for responding to thermal annealing by reducing defect sites and showing slight structural changes as a result of glass relaxation that can alter the electrical properties, the I-V measurements were performed for three independent temperature cycles. The dark and illuminated I-V curves for each sample as a function of temperature for the first thermal cycle are given in Figure 6. In all samples, as the temperature increases, the light and dark currents increase. This indicates that these samples are not metallic, because if they were the current would have decreased with the increase in temperature. This also indicates that the Sn and Pb-doped samples that were estimated from their Raman spectra to have higher concentrations of metal dopant are amorphous semiconductors. Both the C and Si-doped samples exhibit reduced dark and light currents compared to the undoped sample. This is in contrast to the Sn and Pb samples, which have higher currents overall. It should be noted that the dark current for each sample increases significantly above 100 °C. However, the Ge_2_Se_3_ film, even with these dopants, has a glass transition temperature above 300 °C [48]. In fabrication, and in testing, the maximum temperature that the Ge_2_Se_3_ film reached was 140 °C, which is well below the glass transition temperature. Further, the Raman spectra collected before and after annealing at 140 °C reveal no structural changes (Appendix A). At the higher testing temperatures, the extended states conduction mechanism dominates. This is the conduction mechanism that also dominates the dark spectra. The light and dark are therefore expected to be similar as the temperature is increased, since the number of defects (which contributes to light conduction) is reduced in the annealing process.

The most significant changes in the OGT I-V curves occurred between the as-fabricated (no anneal) and the post-first anneal test. Figure 7a shows the pre-anneal and post-first anneal I-V curve collected at room temperature before and after the first temperature cycling measurements. The light and dark spectra show reductions in the current post-annealing, with the most significant change appearing in the undoped sample. However, for both the C and undoped samples, there is a significant increase in the dark current above V_DS_ = 8 V after annealing.

The photocurrent corresponding to the I-V light and dark curves in Figure 7a is given in Figure 7b, presented in a log-log style commonly used to identify regions of conduction, labeled 1, 2, and 3. Region 1 corresponds to the Ohmic conduction region. Region 2 corresponds to the trap-filling conduction region, and region 3 corresponds to the trapped-filled limit region. Figure 8 provides the temperature-dependent I-V curves in the log-log style for all devices during the first temperature cycle.

It can be difficult to identify the onset of trap-filling conduction and the trap-filled limit voltage for many of the sample log-log I-V curves given in Figure 8a,b. However, the trap-filled limit voltage (V_TFL_) can be discerned from the maximum of the Ln(I_ph_/V_DS_) vs. V_DS_ curve; Figure 9 provides an example of the Ln(I_ph_/V_DS_) curves for each sample type on the first thermal cycle (the Appendix A, contains the plots for all thermal cycles). When the photoconductivity is displayed in this way, the peak occurs at the trap-filled limit, and is clearly discernable. This peak occurs at a V_DS_ that decreases as the temperature increases in all cases. The result is that the photoconductivity (I_ph_/V_DS_) increases with temperature at a lower V_DS_. As V_DS_ increases beyond the trap-filled limit, the photoconductivity is independent of temperature. The photoconductivity of the undoped sample during the first anneal cycle is an exception in that there is a slight decrease in photoconductivity with temperature as V_DS_ increases. After the annealing cycle, the undoped sample no longer demonstrates this temperature dependence. Therefore, this anomalous temperature dependence can be attributed to a non-equilibrium in the glass structure with excess trap states (defects), both of which are reduced during the initial temperature anneal. The addition of dopant atoms can produce a glass that is less susceptible to the initial annealing effect due to the dopant atom interactions with the defects present in the glass. However, as the initial annealing effects are irreversible, this is not a concern since the anneal can occur as a fabrication processing step and be avoided on a final product.

The dependence of V_DS_ at the photoconductivity peak, derived from estimating the peaks of each curve in Figure 9, was used to investigate the form of temperature dependence in the relationship between V_DS_ and the peak photoconductivity during the first anneal cycle. As seen in Figure 10a,b, there is a linear dependence of V_DS_ on temperature for each sample type. The slope of the lines corresponds to the negative temperature coefficients of the trap-filled threshold voltages (Figure 10c). This demonstrates that as the temperature increases, the localized or mobility edge states are increasingly filled, resulting in a lower energy requirement for carriers to reach extended states conduction.

The 2nd and 3rd temperature cycles reduce the dependence of V_TFL_ on temperature for all of the samples (Figure 10c). However, this dependence is not eliminated even after the 3rd cycle. In each temperature cycle, the Si-doped device had the largest temperature coefficient of V_TFL_.

It is useful to compare the amplitudes of the photocurrents among the samples; however, the importance of the OGT dark current amplitudes should not be overlooked since the dark current may be critical to the intended OGT application. For example, to function as a selector device for a resistive memory, the dark current should be as low as possible to prevent sneak path currents. Given the increase in dark current as a function of temperature, dark current may be the limited factor in device operation. Figure 11a shows the dark current for each device type on the first anneal cycle. The dark current increases by almost 3 orders of magnitude from 20 to 140 °C in all samples. At higher temperatures, the photocurrent is the same order of magnitude as the dark current, as shown in Figure 11b.

The extended states conduction activation energies were estimated using the dark current measurements at V_DS_ = 7 V using Equation (3). The dark conductivities are plotted as Ln(σ_dark_) vs. 1/kT (Figure 12a) to allow a convenient linear fit to Equation (4) for activation energy (E_a_) determination (the slope of the line fit). The resultant E_a_’s for each thermal cycle are shown in Figure 12b and listed in Table 4.

#### 3.2.2. Responsivity

R is shown in Figure 13 as a function of the LED wavelength. R is calculated using the photocurrent in Equation (2). The responsivity at wavelengths below 500 nm is higher than observed in Si. Additionally, while Si R peaks above 800 nm, the OGT response peaks near 600 nm and maintains a fairly high response between 385 and 1060 nm. This indicates that the OGT gate layer provides a necessary photoresponse that contributes to an enhancement in the underlying Si photoresponse.

#### 3.2.3. Pulse Testing

Each device type was tested using LED light pulses in order to measure the response time. Figure 14 provides the normalized responses of each device type to the LED pulse. The LED pulse was generated using the pulse shown, dashed, at the bottom of the graph.

The rising and falling edges of all OGT device responses to an LED light pulse were modeled using either a single or double exponential function, as shown in Equations (5) and (6). The fit parameters, A and τ, are given in Table 5.

The pulse response rising and falling edge curve fit parameters show that Sn and Pb-doped OGTs have a slower response time compared to the other materials, which may indicate the presence of more trap states. An increase in gap states in Sn and Pb-doped Ge_2_Se_3_ is also supported by the optical energy gap data, as shown in Figure 5 and Table 3. The rising and falling switching times of the C-doped device were the fastest, with times increasing as the doped element size increased (from C to Sn). The heaviest element, Pb, required a double exponential fit for optimum modeling.

## 4. Discussion

The electrical measurements indicate that doping the OGT with group IV elements produces changes in the device electrical properties that could provide useful alternative materials to enable the device to function in a wider range of applications. For example, the intrinsic ‘compliance current’, or the current limiting what the OGT exhibits, can be significantly altered by doping. Figure 6 shows this readily in the case of the C-doped device. This device has a maximum current that is lower than the undoped OGT by more than a factor of two over a wide operating temperature range (up to 100 °C). The C-doped device is a candidate for low-power switching and for device protection in circuits that require low currents.

The speed of the OGT in response to a square light pulse is significantly changed upon doping. While the lower-weight elements of the group (C, Si, and Ge) exhibit somewhat similar switching speeds (within 300 ns), with the switching speeds getting lower as the elements get lighter, the heavier elements, Sn and Pb, are significantly slower. Overall, the switching speed of undoped and Group IV-doped OGTs is nearly five orders of magnitude higher compared to many of the metal oxide semiconductor phototransistors [20], reported single-crystal 2D GeSe-Nanosheets [53], amorphous indium gallium zinc oxide [54,55]-based phototransistors, and organic devices [15,19,21,22,30,32].

The Pb-doped devices have the advantage of showing higher thermal stability with respect to the switching voltages. This is useful in applications that may require a tighter tolerance in response to temperature.

The OGT devices all had similar responsivity curves, differing in the maximum value of R reached, but not in the overall shape of the wavelength response curve. It is anticipated that dopants that are added outside of the Group IV elements will have a more significant effect on the responsivity range; however, this is the subject of a different study. The Raman spectra, activation energy, and Tauc plot-estimated optical energy gap support the change in glass structure due to the doping with group IVA elements, which modified the I_ph_, R, and photoswitching speed of the OGT. For each dopant, there is a noted change in the Raman spectra at the Ge-Ge bonding sites. This is a preferred bond for interacting with the dopants, and the target for influencing optical and electrical changes. Furthermore, the significant alteration in the Raman peak for the Ge-Ge bond site due to Sn and Pb incorporation indicates a clear change in the Ge-Ge structural units within the Ge_2_Se_3_ film. This change could increase responsivity ~2 times more than the undoped OGT at a 470 nm wavelength with a low power density of 0.2 mW/cm^2^ due to a reduction in the optical energy gap, as seen in the Tauc plots for these OGT types. The dark conductivity for the undoped Ge_2_Se_3_ and M + Ge_2_Se_3_ increased exponentially following the Arrhenius equation. This is important for applications in which a low dark current is required to note the increase in dark current with temperature.

It was found that the optical gap, photocurrent, threshold voltage, and speed were altered depending upon the dopant added, with the heavier elements, Sn and Pb, producing the greatest change in these parameters. The lightest element, C, was similar to the undoped control, but displayed switching times that were approximately 300 ns faster and a photocurrent almost twice as low as the undoped OGT. Table 6 summarizes the OGTs as a function of the dopant type and ranks the changes by 1 (lowest) to 5 (highest).

In looking at the parameter summary in Table 6, it is notable that the largest dopant element, Pb, produces the most polar responses: either the ‘highest’ or the ‘lowest’ of a given parameter. For example, Pb-doped OGTs are the slowest and have the highest dark current, but yet they have the highest photocurrent and highest IR-absorbing capacity. If an application required a higher threshold voltage, with minimal temperature fluctuations, the Pb-doped OGT would be the best choice, having the highest threshold voltage and the lowest threshold voltage temperature coefficient. This OGT would be useful in an application where speed is not as important as thermal stability. On the other hand, the lightest element, C, produces a fast switching device compared to the undoped OGT, with a high UV absorption and low dark current. It also has a low threshold voltage temperature coefficient. However, it also has a low photocurrent. In an application where speed, a low dark current, and the thermal stability of the threshold voltage were important, this would be the better choice.

## 5. Conclusions

Group IVA element-doped OGTs were fabricated and underwent electrical characterization as a function of temperature in order to evaluate the effect the Ge-Ge bond has on OGT operating parameters. The major differences in functional parameters such as the speed, threshold voltage, dark current and photocurrent were measured. Work is currently underway to study the operating mechanisms of the OGT under the influence of dopants, and its potential for charge storage.

## Figures and Tables

**Figure 1 micromachines-15-01000-f001:**
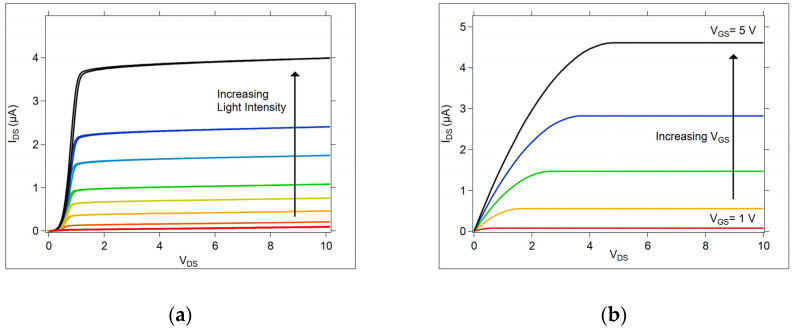
The OGT and MOSFET I-V curves. (**a**) An OGT device produces a current with an amplitude dependent upon the intensity of incident light on the gate; (**b**) a family of I-V curves for a MOSFET showing the current amplitude dependence on the voltage applied to the gate.

**Figure 2 micromachines-15-01000-f002:**
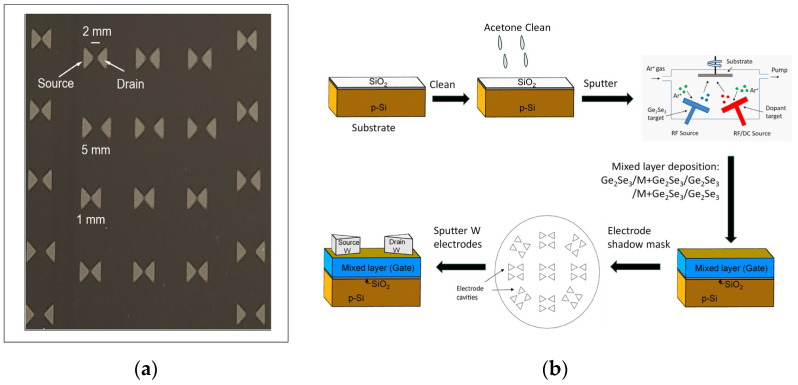
An image of completed OGTs and OGT fabrication steps. (**a**) Top-down photograph of completed OGTs; (**b**) Schematic diagram of the fabrication process for the doped OGT device, comprising alternating stack layers of Ge_2_Se_3_ (approx. 100 Å) and M + Ge_2_Se_3_ (approx. 10 Å) on a p-Si/SiO_2_. In all cases, the OGT gate achieves an overall thickness of ~320 Å. The undoped OGT is fabricated without the co-sputtered layers, to a total thickness of ~320 Å.

**Figure 3 micromachines-15-01000-f003:**
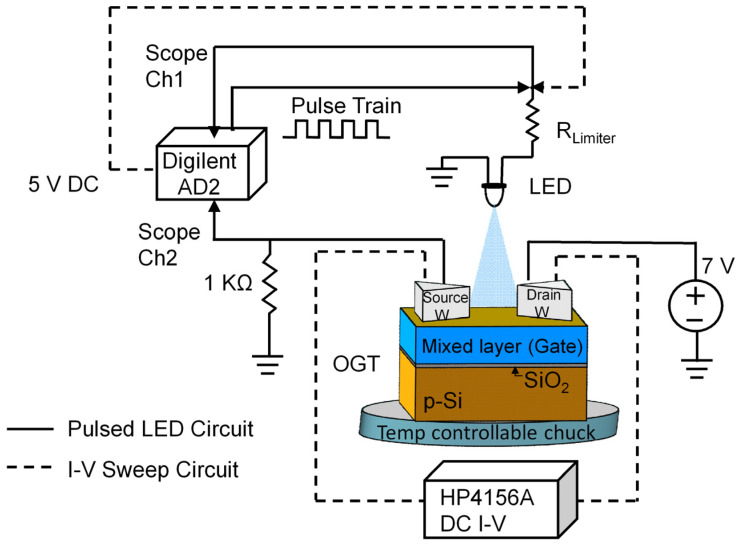
Electrical setup for OGT pulse testing and I-V sweep measurement. The equipment connectivity for pulse testing corresponds to the solid line connections, and I-V sweeps correspond to the dashed line connections. The + drain voltage was used for the C, Si, undoped, and Sn-doped devices and the drain voltage was used for the Pb device.

**Figure 4 micromachines-15-01000-f004:**
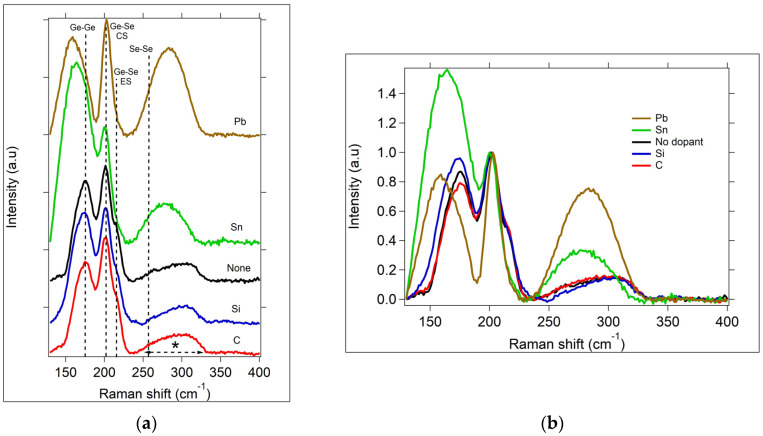
(**a**) Raman spectra normalized at 200 cm^−1^ (Ge-Se CS peak) for Ge_2_Se_3_ and M + Ge_2_Se_3_, samples with M = C, Si, Sn, or Pb. Dashed vertical lines correspond to the Raman shifts expected for the a-Ge_x_Se_100-x_ material: Ge-Ge bond; Ge-Se corner-sharing (CS) tetrahedra; Ge-Se edge-sharing (ES) tetrahedra; Se-Se bonds (no peak present in the data); and * = additional region of Ge-Ge bonding. (**b**) Overlapping normalized Raman spectra from (**a**).

**Figure 5 micromachines-15-01000-f005:**
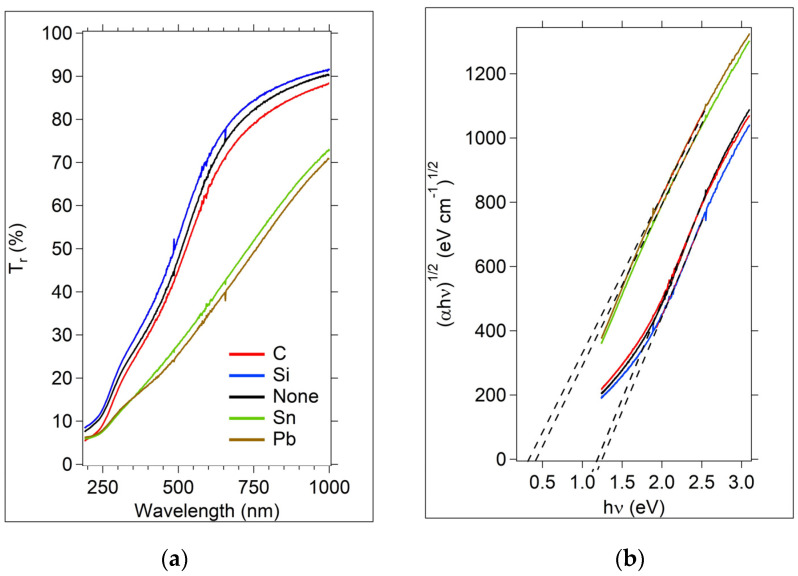
UV-Vis transmission and Tauc plots for the Group IVA samples, Ge_2_Se_3_ and M + Ge_2_Se_3_. (**a**) %transmission data (T_r_); (**b**) Tauc plots obtained from the data in (**a**). The legend for both graphs is in (**a**).

**Figure 6 micromachines-15-01000-f006:**
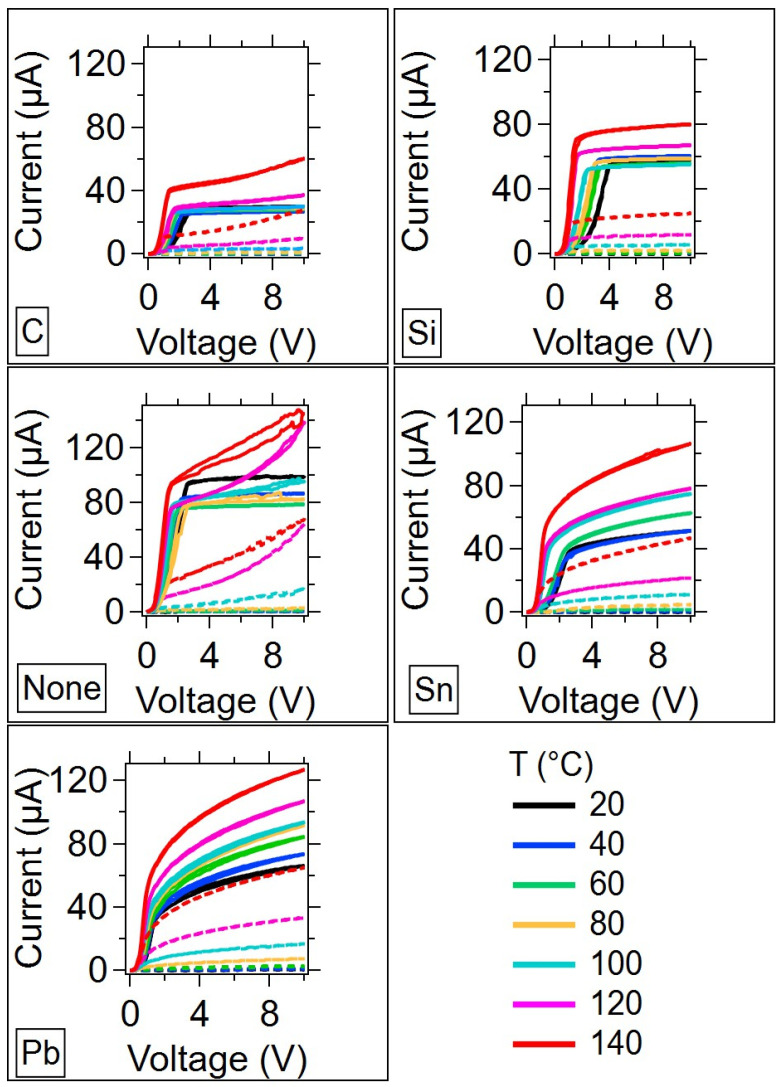
I-V curves for each sample as a function of temperature during the first temperature cycle. The voltage corresponds to the drain to source voltage, V_DS_. Solid lines = light; Dashed lines = dark.

**Figure 7 micromachines-15-01000-f007:**
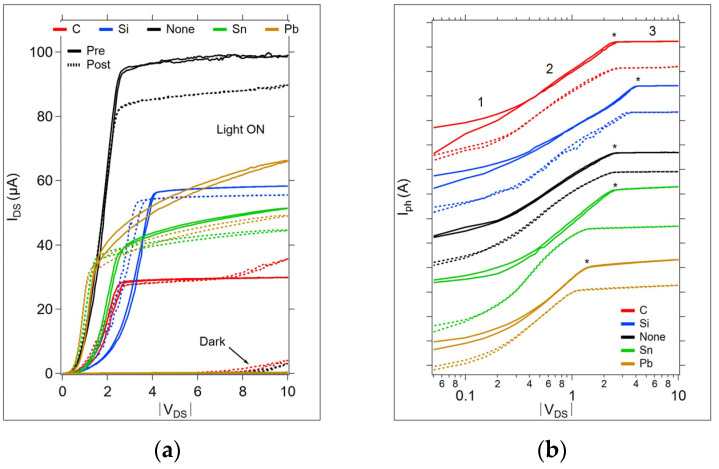
Pre- and post-anneal I-V data collected at room temperature. (**a**) Light and dark traces for each sample before any thermal cycling (Pre) and after the first thermal cycle to 140 °C (Post). (**b**) Waterfall format log-log plot comparing each I-V curve. Common designations for the conduction regions are given by 1 (Ohmic), 2 (trap-filling region), and 3 (trap-filled region). * denotes the trap-filled limit voltage, V_TFL_, on the pre-anneal cycle.

**Figure 8 micromachines-15-01000-f008:**
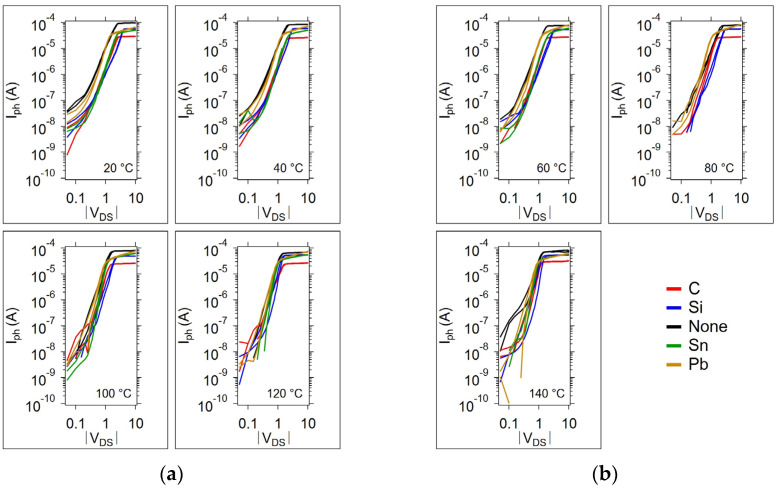
Photocurrent as a function of the drain to source voltage |V_DS_| and temperature for all samples during the first temperature cycle. (**a**) 20, 40, 100, and 120 °C; (**b**) 60, 80, and 140 °C.

**Figure 9 micromachines-15-01000-f009:**
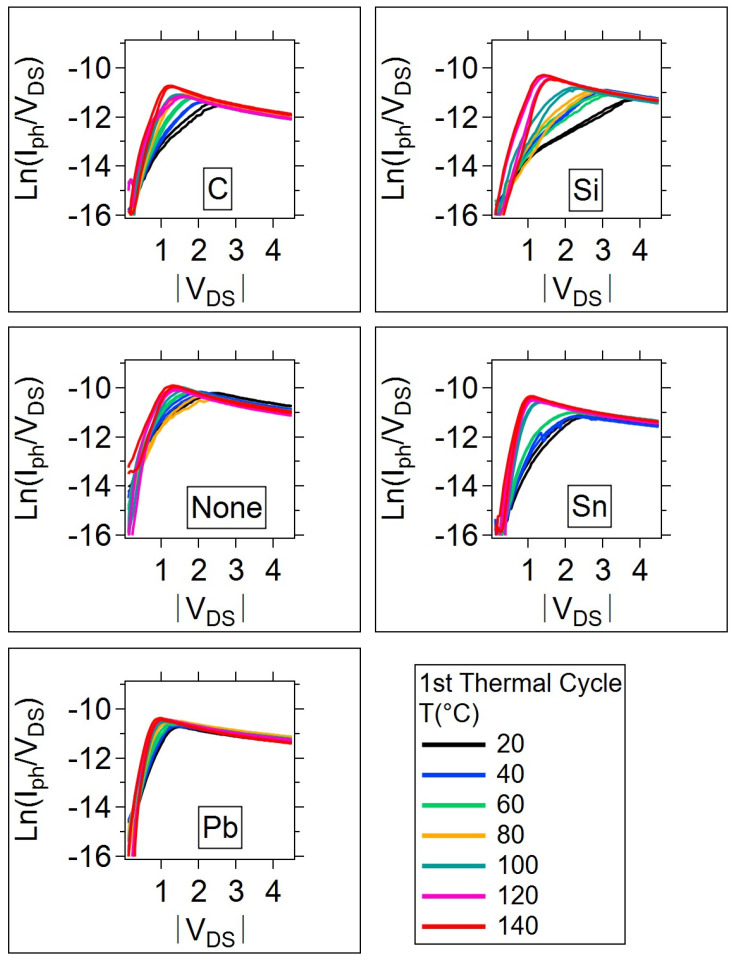
Natural log of the photoconductivity, Ln(I_ph_/V_DS_), for the 1st thermal cycle as a function of the drain to source voltage |V_DS_|, and temperature.

**Figure 10 micromachines-15-01000-f010:**
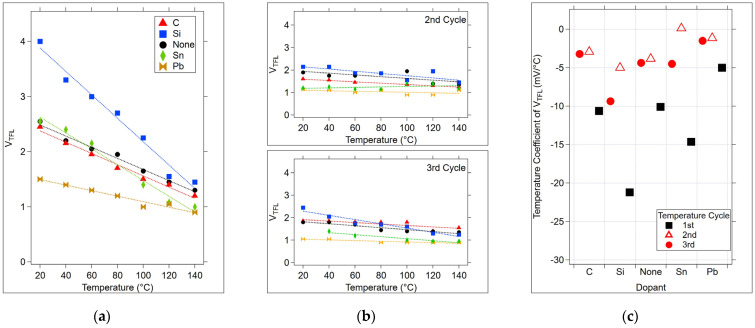
The relationship between V_DS_ at the photoconductivity peak in Figure 9 and the temperature for each sample. (**a**) V_TFL_ for the 1st temperature cycle, derived from Figure 9. (**b**) V_TFL_ for the 2nd and 3rd temperature cycles (Appendix A). (**c**) Temperature coefficients (the slopes of the lines in (**a**,**b**)) for V_TFL_, for each sample. In (**a**,**b**), dashed lines correspond to the linear fit for each sample.

**Figure 11 micromachines-15-01000-f011:**
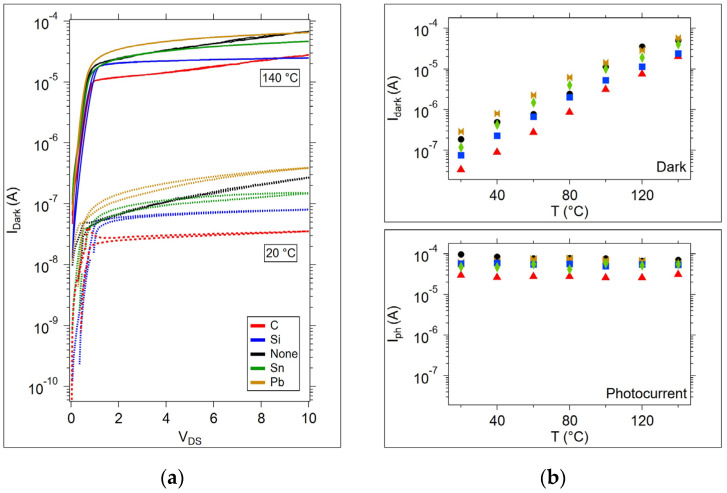
Dark currents and photocurrents during the first thermal cycle. (**a**) Dark currents at T = 20 and 140 °C. Dashed lines correspond to T = 20 °C. (**b**) Dark current and photocurrents for each sample type over the entire temperature range measured at V_DS_ = 7 V.

**Figure 12 micromachines-15-01000-f012:**
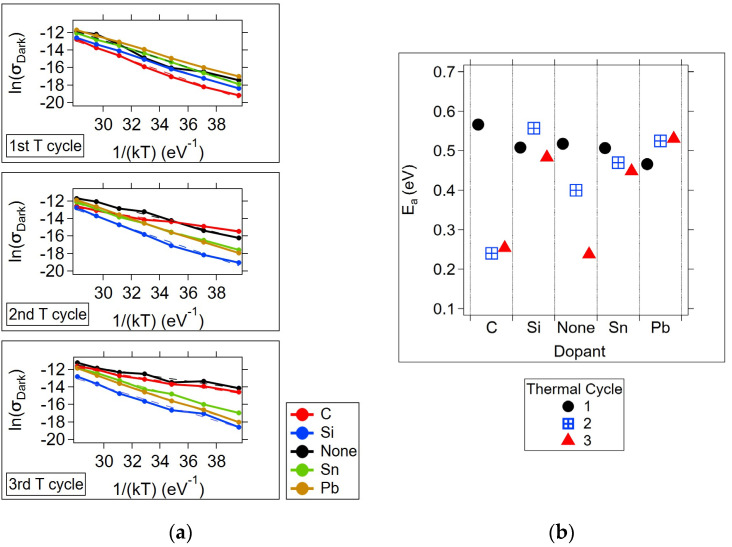
Dark activation energy calculations for extended states conduction. (**a**) Dark conductivity (calculated at V_DS_ = 7 V) vs. 1/kT for each thermal cycle. The calculation of the activation energy was performed using Equation (4). Dashed lines correspond to linear fits. (**b**) Activation energy, E_a_, for each dopant and thermal cycle.

**Figure 13 micromachines-15-01000-f013:**
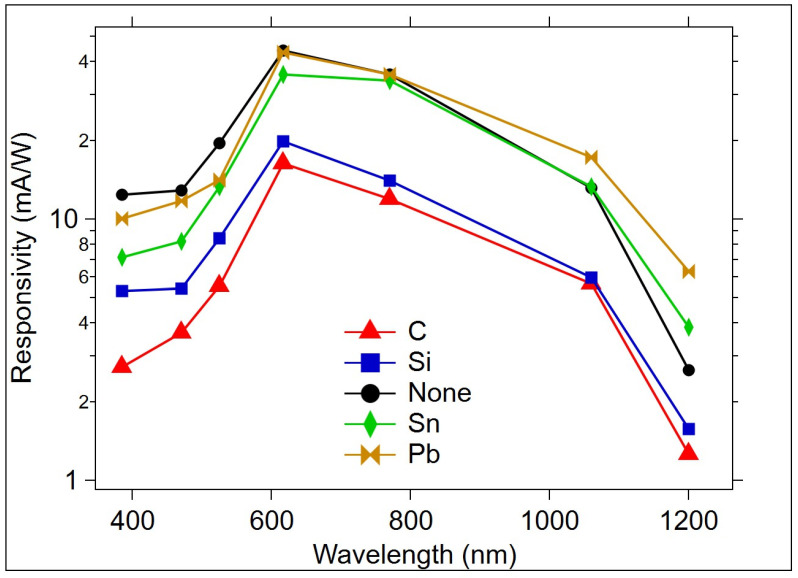
Responsivity as a function of wavelength. |V_DS_|= 7 V.

**Figure 14 micromachines-15-01000-f014:**
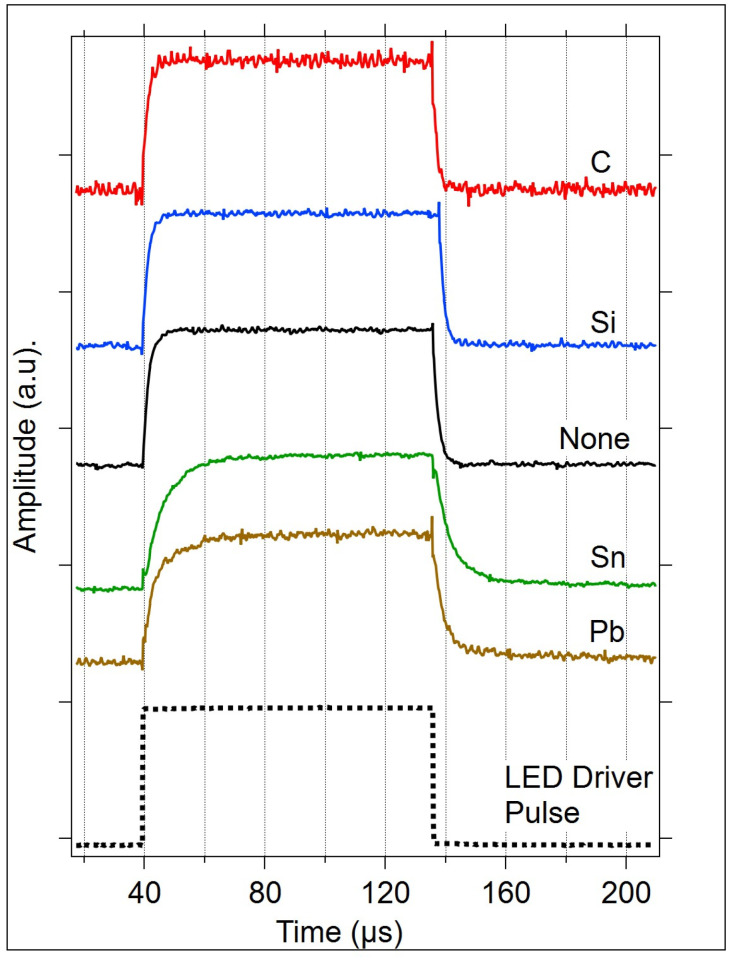
Electrical response of each OGT type to a light pulse. The LED was driven by the pulse shown (dashed). All pulses are normalized to 1 for ease of comparison.

**Table 1 micromachines-15-01000-t001:** LED wavelength, part, vendor and LED power density.

Wavelength (nm)	Part, Vendor	Power Density (mW/cm^2^)
385	VAOL-5GUV8T4, Visual Communication Company	7.5
470	C503B, Cree	8.5 ^a^
525	HLMP-CM1A-560DD, AvagoTechnology	7.3
616	TLCY5800-ND, Vishay Semiconductors	2.1
770	MTE1077N1-R, Marktech Optoelectronics	7.0
1060	MT51060-IR, Marktech Optoelectronics	3.8
1200	MTE0012-525-IR, Marktech Optoelectronics	0.35

^a^ In the temperature studies, the power density was 9.8 mW/cm^2^.

**Table 2 micromachines-15-01000-t002:** Summary of the electrical characterization tests.

Measurement Group	Metric	Wavelength (nm)	V_DS_ (V)
I-V measurements	I_DS_ vs. V_DS_	See Table 1	0 ⇄ 10
	R vs. Wavelength	See Table 1	7
Temperature-dependent I-V 20 → 140 °C, ∆T = 20 °C.	I_DS_ vs. V_DS_; ln(I_DS_) vs. V_DS_; ln(I_ph_) vs. V_DS_	470	0 ⇄ 10
	(1) ln (I_DS_/V_DS_) vs. 1/T; (2) Extended states conduction E_a_ vs. dopant.	470	7
Pulsed LED tests	V_DS_ vs. Time	470	7

**Table 3 micromachines-15-01000-t003:** M + Ge_2_Se_3_ OGT optical energy gap estimate from the Tauc plot in Figure 5.

M	$E_{0}$ (eV)
C	1.19
Si	1.23
None	1.27
Sn	0.403
Pb	0.392

**Table 4 micromachines-15-01000-t004:** Estimated dark conductivity activation energy E_a_ for each temperature cycle.

	Cycle 1	Cycle 2	Cycle 3
Dopant	E_a1_ (eV)	E_a2_ (eV)	E_a3_ (eV)
C	0.566	0.24	0.253
Si	0.508	0.557	0.483
None	0.518	0.4	0.237
Sn	0.507	0.47	0.448
Pb	0.466	0.525	0.53

Note: All goodness of fit parameters, r^2^, were >0.99 except for the undoped sample, which had r^2^ = 0.96 for cycle 1 and 0.97 for cycle 3.

**Table 5 micromachines-15-01000-t005:** Pulse switching response exponential equation model fit parameters.

	Rising Edge		Falling Edge	
Dopant	τ (µs)	A	τ (µs)	A
C	1.66	−0.041	1.55	0.034
Si	1.80	−0.086	1.66	0.075
None	1.90	−0.12	1.75	0.014
Sn	7.2	−0.073	5.80	0.068
Pb *	2.51/11.3	−0.047/−0.029	3.19/32.33	0.069/0.006

* Pb was fit to a double exponential, Equation (6).

**Table 6 micromachines-15-01000-t006:** OGT parameters tuned by dopant. 1 = lowest; 5 = highest value of given parameter.

Dopant	Speed	V_TH_ *	V_TH_ * T coeff	IR abs.	UV abs.	E_o_	I_dark_	I_ph_
C	5	3	2	1	5	3	1	1
Si	4	1	5	2	3	4	2	4
None	3	2	4	3	4	5	4	3
Sn	2	4	3	4	2	2	3	2
Pb *	1	5	1	5	1	1	5	5

* Starting at the 2nd anneal, and assigning threshold voltage at the trap-filled limit value.

## Data Availability

The data presented in this study are available upon request from the corresponding author.

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
