# Peer review of "Effects of Group IVA Elements on the Electrical Response of a Ge_2_Se_3_-Based Optically Gated Transistor"

_micromachines, 2024, doi:10.3390/mi15081000_

Round 1
Reviewer 1 Report
Comments and Suggestions for Authors
In this manuscript, authors adopt the strategy of doping IVA group elements C, Si, Sn and Pb in the a-Ge2Se3 gate materials of phototransistors to explore the effects of the changes of Ge-Se and Ge-Ge bonds on threshold voltage, photoresponse and photocurrent.,This work can be considered for publication after the following issues.
1. Alternately stacked gate materials are shown in the fabrication process in figure 2. The effect of multi-layer alternating stacking on devices is not discussed in this paper. Why?
2. Use the transmission electron microscope to characterize the gate material layer to verify the presence of multiple layers in the device.
3. In your introduction and discussion, there is insufficient explanation about phototransistors. It is recommended to quote the following relevant documents (DOI: 10.1039/d2tc04340h, DOI:10.1002/adfm.202113053, DOI:10.1109/TED.2020.3001492, DOI:10.1007/s42765-023-00318-z, DOI:10.1002/aelm.201901255) to provide more background support for your research, which will help readers better understand your research contribution and background.
4. The typesetting of the table in figure 9 is very poor, the information in the figure cannot be clearly seen, and the content of figure 9 is not described much in the article, so it is recommended to put the content of figure 9 in the supporting information of the article.
Reviewer 2 Report
Comments and Suggestions for Authors
Please see that attached PDF

Round 2
Reviewer 1 Report
Comments and Suggestions for Authors
The authors have answered all the questions and made the relative revisions.
Comments on the Quality of English LanguageThe English can be refined.
Reviewer 2 Report
Comments and Suggestions for Authors
I very much appreciate authors efforts in performing complementary experiments to reinforce the conclusions made in the original version. The added text further clarifies the takeaway message.
I am happy to encourage publication of this study.